# Differential Effects of Human Immunodeficiency Virus Nef Variants on Pulmonary Vascular Endothelial Cell Dysfunction

**DOI:** 10.3390/idr17030065

**Published:** 2025-06-06

**Authors:** Amanda K. Garcia, Noelia C. Lujea, Javaria Baig, Eli Heath, Minh T. Nguyen, Mario Rodriguez, Preston Campbell, Isabel Castro Piedras, Edu Suarez Martinez, Sharilyn Almodovar

**Affiliations:** 1Department of Immunology & Molecular Microbiology, School of Medicine, Texas Tech University Health Sciences Center, Lubbock, TX 79430, USA; amandakristene.garcia@ttuhsc.edu (A.K.G.);; 2Department of Chemical Engineering, Edward E. Whitacre Jr. College of Engineering, Texas Tech University, Lubbock, TX 79430, USA; 3College of Arts and Sciences, Texas Tech University, Lubbock, TX 79430, USA; 4Center for Biotechnology & Genomics, Texas Tech University, Lubbock, TX 79430, USAisabel.castro@ttu.edu (I.C.P.); 5Department of Biology, University of Puerto Rico at Ponce, Ponce, PR 00732, USA; edu.suarez@upr.edu; 6Center for Tropical Medicine & Infectious Diseases, School of Medicine, Texas Tech University Health Sciences Center, Lubbock, TX 79430, USA

**Keywords:** HIV proteins, endothelial cell dysfunction, pulmonary vascular remodeling, HIV Nef, *ICAM1*, *VCAM1*, E-selectin, apoptosis, endothelin-1, inflammation, eNOS, pulmonary vascular cell co-culture

## Abstract

**Background:** Human Immunodeficiency Virus (HIV) infections remain a source of cardiopulmonary complications among people receiving antiretroviral therapy. Still to this day, pulmonary hypertension (PH) severely affects the prognosis in this patient population. The persistent expression of HIV proteins, even during viral suppression, has been implicated in vascular dysfunction; however, little is known about the specific effects of these proteins on the pulmonary vasculature. This study investigates the impact of Nef variants derived from HIV-positive pulmonary hypertensive and normotensive donors on pulmonary vascular cells in vitro. **Methods:** We utilized well-characterized Nef molecular constructs to examine their effects on cell adhesion molecule gene expression (*ICAM1*, *VCAM1*, and *SELE*), pro-apoptotic gene expression (*BAX*, *BAK*), and vasoconstrictive endothelin-1 (*EDN1*) gene expression in endothelial nitric oxide synthase (eNOS) nitric oxide and the production and secretion of pro-inflammatory cytokines over 24, 48, and 72 h post-transfections with Nef variants. **Results:** HIV Nef variants SF2, NA7, and PH-associated Fr17 and 3236 induced a significant increase in adhesion molecule gene expression of *ICAM1*, *VCAM1*, and *SELE*. Pulmonary normotensive Nef 1138 decreased *ICAM1* gene expression, but had increased *VCAM1*. PH Nef ItVR showed a consistent decrease in *ICAM1* and no changes in *SELE* and *VCAM1* expression. Further gene expression analyses of pro-apoptotic genes *BAX* and *BAK* demonstrated that Nef NA7, SF2, normotensive Nef 1138, and PH Nef Fr8, Fr9, Fr17, and 3236 variants significantly increased gene expression for apoptosis. Normotensive Nef 1138, as well as PH Nef Fr9 and ItVR, all displayed a statistically significant decrease in *BAX* expression. The expression of *EDN1* had a statistically significant increase in samples treated with Nef NA7, SF2, normotensive Nef 2044 and PH Nef 3236, Fr17, and Fr8. Notably, PH-associated Nef variants sustained pro-inflammatory cytokine production, including IL-2, IL-4, and TNFα, while anti-inflammatory cytokine levels remained insufficient. Furthermore, eNOS was transiently upregulated by all Nef variants except for normotensive Nef 2044. **Conclusions:** The distinct effects of Nef variants on pulmonary vascular cell biology highlight the complex interplay between Nef, host factors, and vascular pathogenesis according to the variants.

## 1. Introduction

Current antiretroviral therapy has revolutionized the clinical management of HIV by promoting a continuous decrease in all-cause mortality in persons living with HIV (PLWH) [1,2,3]. The recipe for this success is that antiretroviral therapy suppresses viral replication, allowing for the reconstitution of the immune system and decreased incidence of opportunistic infections that define AIDS. Nevertheless, until a definite cure is achieved, PLWH face significant challenges, including poor adherence to antiretroviral drug regimens, development of drug-resistance mutations, uninterrupted viral evolution that continues even during viral suppression, reactivation of viral reservoirs, and additional immune challenges introduced by co-infections. In addition, PLWH remains at the edge of developing several comorbidities that affect the function of several end organs, including the cardiovascular-pulmonary system. PLWH have significantly increased the odds of developing several lung and pulmonary diseases, including asthma, COPD, pulmonary fibrosis, lung cancer, and pulmonary hypertension (PH). Among these, PH is a severe cardiopulmonary vascular complication of HIV infection with a poor prognosis that leads to right heart failure and death [4,5,6]. Globally, HIV-associated PH (HIV-PH) has been reported to affect between 0.5 and 14% of HIV+ study cohorts [5,7,8,9,10,11,12,13]. A recent systematic review and meta-analysis reported a mean survival rate of 37 months and a hospitalization rate of 71%, with significant income and regional disparities [14]. Upper-middle-income countries had a three-year survival rate of 80.5%, while lower-middle-income countries had a three-year survival rate of 23.9%. Geographically, this review also reveals that European studies had the highest one-year survival rate at 91.6%. This contrasts with American studies having a one-year survival rate of 79.5 and African studies at 35.1%.

How HIV contributes to these diseases remains largely unknown. One longstanding hypothesis is that the persistent expression of HIV proteins, including Nef, gp120, and Tat, in patients treated with antiretroviral therapy may contribute to HIV comorbidities [15,16]. For instance, HIV Nef is a multifunctional accessory protein and a primary determinant of vascular pathogenesis that persists in the lungs of PLWH treated with antiretroviral therapy, even in the absence of viral replication [17,18]. Nef interacts with host cell proteins and intersects molecular processes and signaling pathways from the lipid rafts to the sub-molecular level in the cellular organelles [19,20]. Moreover, Nef travels extracellularly in vesicles [17,21,22,23,24], which perpetuates its paracrine effects. Previous studies identified HIV Nef protein transfer to the endothelium to cause vascular dysfunction [25]. Separate studies also discovered Nef within plexiform lesions in rhesus macaques infected with a chimeric simian–human immunodeficiency virus [26]. In subsequent studies, we identified, sequenced, and sub-cloned Nef polymorphisms that were over-represented in HIV+ pulmonary hypertensive donors, compared to their normotensive counterparts [27]. Such isolates represent polymorphisms that were shared between HIV-PH individuals and macaques with PH-like disease: histidine to tyrosine at position 40 (H_40_Y) and alanine to proline at position 53 (A_53_P). We also communicated specific mutations in Nef functional domains, including polymorphisms near M_20_, whose functional role is to interact with the adaptor protein 1 (AP-1) and efficiently prevent MHC-1 trafficking to the membrane [28], a L_58_V mutation in the CD4 down-regulation domain [29], an E_63_G mutation in the acidic cluster mediating the sequestration of MHC-1 in the *trans*-Golgi network [30], a cluster of mutations in the M_79_I/T_80_N/Y_81_F phosphorylation site for protein kinase C [31], and substitutions within the PxxP motif (proline-rich area essential for Nef interaction with SH3 domain-containing proteins [32], where x is any amino acid). Herein, we utilized banked Nef molecular constructs derived from HIV pulmonary hypertensive and normotensive donors, along with the well-characterized NA7- and SF2-*nef* isolates, as additional sources of Nef variations to investigate the impact of Nef variants in different aspects of pulmonary vascular endothelial cell biology and dysfunction.

## 2. Materials and Methods

### 2.1. Cell Culture

Human pulmonary artery smooth muscle cells and human pulmonary artery endothelial cells from uninfected donors were purchased from Lonza (Walkersville, MD, USA) and used at 60–80% confluency between passages 7–10 and 6–7, respectively. In this study, we utilize an endothelial/smooth muscle cell vascular co-culture model to provide a more physiologically relevant context for investigating pulmonary vascular cell responses to HIV Nef. Specifically, endothelial and smooth muscle cells were seeded at a 1.8:1 ratio in 6-well tissue culture plates with the corresponding smooth muscle cell growth medium (SmBM supplemented with SmGM-2, Lonza) mixed with endothelial cell growth medium (EBM-2 supplemented with EGM-2, Lonza). Cell cultures were maintained in a 37 °C humidified incubator at 5% CO_2_.

### 2.2. Transfections of Vascular Cells with HIV Nef Constructs

HIV *nef* isolates associated with pulmonary hypertension were previously subcloned into the HaloTag vector system (Promega, WI, USA), followed by the functional characterization of their ability to downregulate CD4, a canonical function of Nef [27]. This study includes the following Nef constructs: 1138 and 2044 (derived from two HIV-positive normotensive donors), Fr8, Fr9, Fr17, 3236, and ITVR (derived from four HIV-positive pulmonary hypertensive donors). In addition, we included SF2 and NA7 to represent additional variants of the Nef protein. We used an empty vector (BL1, vector without *nef* coding region) as an internal negative control. Pulmonary vascular co-cultures consisting of pulmonary artery endothelial and smooth muscle cells were seeded on 6-well plates at a density of 64,700 cells/cm^2^ and allowed to adhere overnight at 37 °C, 5% CO_2_. Cells were at 60–70% confluency before transfections with HIV *nef* plasmids. Cells were transfected with 2.5 micrograms of each Nef construct using Lipofectamine 3000 Transfection Reagent (L3000015, Invitrogen, Waltham, MA, USA), following the manufacturer’s protocol. Cells were harvested after 24, 48, or 72 h to assess the molecular endpoints described below.

### 2.3. Gene Expression Analyses

Cells were harvested for total RNA extraction at 24, 48, and 72 h post-Nef transfections. Cells were lysed using an Aurum Total RNA Kit (7326820, Bio-Rad, Hercules, CA, USA). Total RNA was treated with DNase I to remove genomic DNA. The concentration and quality of the extracted RNA solutions were determined using spectrophotometry (NanoDrop ND-2000, Thermo Fisher Scientific, Waltham, MA, USA). Samples were reverse-transcribed into cDNA using the RT2 First Strand Synthesis Kit (330401, Qiagen, Germantown, MD, USA). The expression of *BAX*, *BAK, ICAM1*, *VCAM1*, *SELE*, and *EDN1* genes were measured by quantitative PCR using a CFX96 Touch Deep Well Real-Time PCR System (Bio-Rad), with Power Track SYBR green qPCR master mix (A46109, Applied Biosystems, Waltham, MA, USA) and primers indicated in **Table 1**. We used the 2^−∆∆C^T method to quantify mRNA transcripts normalized to ribosomal protein lateral stalk subunit P0 (RPLP0) as an internal reference gene. Statistical analyses were performed using GraphPad Prism software (GraphPad Software, Inc., Boston, MA, USA). One-way ANOVA and Tukey’s Multiple Comparisons Test were used to compare fold changes.

### 2.4. Quantitation of Inflammatory Cytokines

Cell culture supernatants of Nef-transfected cells were collected at 24, 48, or 72 h and stored at −80 °C until analysis. We used the Meso Scale Discovery V-PLEX Pro-Inflammatory Panel I (K15049D, Meso Scale Diagnostics, Rockville, MD, USA) to measure IFN-γ, IL-1β, IL-2, IL-4, IL-6, IL-8, IL-10, IL-12p70, IL-13, and TNF-α, according to the manufacturer’s instructions. Briefly, samples and standards were added to pre-coated plates and incubated for 2 h. Plates were then washed and read on a MESO QuickPlex SQ120 imager (Meso Scale Diagnostics, Rockville, MD, USA) to quantify electrochemiluminescence. The cytokine concentrations were determined using MSD Discovery Workbench Software (Meso Scale Diagnostics, Rockville, MD, USA). Statistical analyses were performed using one-way Welch ANOVA and Brown–Forsythe tests (not assuming equal standard deviations) and unpaired T tests with Welch’s correction, with individual variances computed for each comparison using GraphPad Prism software version 10 for MacOS (Boston, MA, USA).

### 2.5. Endothelial Nitric Oxide Synthase (eNOS) Measurement

The levels of eNOS in cell culture supernatants were measured using a Human eNOS/NOS3 ELISA kit (EH169RB, Invitrogen, Waltham, MA, USA). Briefly, samples were diluted 2-fold, and standards loaded onto the plate were incubated overnight at 4 °C on a horizontal orbital microplate shaker. The wells were washed four times with the supplied wash buffer. Then, samples were incubated in biotin-conjugated solution for 1 h at room temperature with shaking, followed by a 45 min incubation with streptavidin solution at room temperature. After a washing step, TMB substrate was added to each well and incubated for 30 min at room temperature in the dark, followed by the addition of stop solution to each well. The absorbance at 450 nm was read within 30 min using a TECAN infinity M1000 microplate reader (Tecan Austria GmnH, Untersbergstr, Austria).

### 2.6. Pulmonary Vascular Cell Co-Culture and Proliferation Monitoring

The growth of co-cultured HPAEC and PASMC cell populations was monitored over a 72-h period following plating. To achieve this, stable cell-tracking fluorescent dyes were used. Briefly, cells were trypsinized, counted, and resuspended at the desired concentration. HPAECs were labeled with CellTracker Green CMFDA (Cat #C7025, Invitrogen, Waltham, MA, USA), while PASMCs were labeled with CellTracker Red CMTPX (Cat #C34552, Invitrogen, Waltham, MA, USA). Labeling was performed by incubating the cells in suspension with 25 µM of the corresponding dye for 45 min at 37 °C in a humidified atmosphere containing 5% CO_2_. After labeling, cells were seeded together at a ratio of 1.8:1 (HPAEC:PASMC). Cells were monitored over 72 h using the CellCyte (Cellcyte X, Cytena, San Diego, CA, USA) imaging system, with images acquired every 3 h. Data were obtained from two independent replicates.

### 2.7. Western Blot Analysis

Cells were lysed using RIPA buffer supplemented with protease inhibitors. Protein concentration was determined using a DC Protein Assay (5000113-5, Bio-Rad, Hercules, CA, USA) according to the manufacturer’s instructions, measuring absorbance at 750 nm. Protein samples were denatured by boiling for 5 min in sample loading buffer. Equal amounts of protein were resolved using electrophoresis on 10% SDS-PAGE gels and transferred to PVDF membranes (5000111, Bio-Rad, Hercules, CA, USA) using a Trans-Blot Turbo Transfer System (1704150, Bio-Rad, Hercules, CA, USA). Membranes were blocked for 1 h at room temperature in 5% BSA. After blocking, membranes were incubated overnight at 4 °C, with primary antibodies diluted in TBST supplemented with 5% BSA. The following antibodies and dilutions were used: anti-Halo Tag (1:1000, Cat #G9211, Promega, WI, USA), anti-EDN1 (1:1000, Cat #A0686, Abclonal, MA, USA), anti-ICAM1 (1:2000, Cat #A5597, Abclonal, MA, USA), and anti-Vinculin (1:10,000, Cat #ab129002, Abcam, MA, USA), which was used as a loading control. Following primary incubation, membranes were washed and incubated for 2 h with HRP-conjugated secondary antibodies: anti-rabbit IgG (1:10,000, Cat #7074S, Cell Signaling, MA, USA) and anti-mouse IgG (1:10,000, Cat #W402B, Promega, WI, USA). Signal detection was performed using a SuperSignal West Pico PLUS (Thermo Fisher Scientific, Waltham, MA, USA), and chemiluminescent images were acquired using an Azure Imaging System (c300, Azure Biosystems, Dublin, CA, USA).

## 3. Results

### 3.1. Pulmonary Vascular Direct Culture System

In this study, we utilized pulmonary artery endothelial and smooth muscle cells in co-culture to investigate pulmonary vascular cell responses to HIV Nef. In general, smooth muscle cells have higher proliferation rates than endothelial cells, which lead to significant smooth muscle cell overgrowth in co-culture models. To account for this, we started our vascular co-cultures with HPAEC: PASMC seeded at a 1.8:1 ratio. Each cell type was labeled with cell-permeable dyes for cell population tracking over 72 h (**Figure 1A**). We observed that the endothelial cell population remained fairly stable over the 72 h assays; however, smooth muscle cells demonstrated a progressive increase, ultimately reaching a three-fold excess over endothelial cells (**Figure 1B**). We confirmed the physiological relevance of these observations by quantifying endothelial and smooth muscle cells in normal pulmonary arteries. According to our results, smooth muscle cells can be found at a 3.4:1 ratio with endothelial cells (**Figure 1C**). Together, these results suggest that our in vitro model allows for physiologically relevant representation of the endothelial and smooth muscle cell populations essential for our pulmonary vascular Nef studies.

### 3.2. Effects of Co-Culture Model on Cell Adhesion Molecules

To examine what effect the co-culture model had on the baseline expression of typical markers for endothelial dysfunction, we measured the gene expression of *ICAM1* and *SELE* in endothelial cell monocultures compared to endothelial and smooth muscle cell co-cultures, which were used throughout the study. Our findings agreed with previous studies that have examined the expression of cell adhesion molecules in EC monocultures compared to EC and SMC co-cultures, and found cell adhesion molecules to be elevated in EC monocultures [33]. We found that, when compared to EC-SMC co-cultures, EC only monocultures had a statistically significant increase in baseline *ICAM1* expression (*p* ≤ 0.0001) (**Figure 2**). The same was true for *SELE* (*p* ≤ 0.0001). Since our results align with those in the literature, this work focuses on the impact of HIV Nef on pulmonary cell biology using direct coculture systems. 

### 3.3. HIV Nef Impairs Pulmonary Vascular Cell Activation

HIV Nef promotes vascular inflammation, in part by upregulating the expression of adhesion molecules like intercellular adhesion molecule-1 (ICAM1), vascular adhesion molecule-1 (VCAM1), and endothelial leukocyte adhesion molecule 1 (E-selectin) by the vascular endothelium. Vascular endothelial activation by oxidative stress, pro-inflammatory cytokines, or infections is characterized by increased expression of ICAM1, VCAM1, and E-selectin to facilitate leukocyte adhesion. Thus, the participation of these adhesion molecules is critical during the initiation and perpetuation of vascular inflammatory processes. In this experiment, we measured the expression of the *ICAM1*, *VCAM1*, and *SELE* genes by quantitative PCR at 24, 48, and 72 h post-Nef transfections. We confirmed Nef protein expression post-transfection using Western blot (**Figure 1D**). The expression of Nef was higher at 24 h compared to later timepoints, and had variable Nef expression among the *nef* constructs.

Next, we assessed the gene and/or protein expression of cell activation markers. Our results show statistically significant differences in *ICAM1* expression between some of the Nef treatments and the control following Nef transfections (**Figure 3**). At 24 h after Nef transfections, NA7 (*p* ≤ 0.05) and ItVR (*p* ≤ 0.01) showed a significant decrease in *ICAM1* gene expression compared to the BL1 control, while SF2 (*p* ≤ 0.001) and 3236 (*p* ≤ 0.001) showed an increase. Interestingly, at 48 h post-transfection, SF2 (*p* ≤ 0.001), Fr8 (*p* ≤ 0.05), Fr17 (*p* ≤ 0.0001), and 3236 (*p* ≤ 0.001) showed a significant increase in *ICAM1* gene expression, whereas ItVR showed a significant decrease compared to the control at the same time point. At a later time point, 72 h, NA7 (*p* ≤ 0.0001), SF2 (*p* ≤ 0.01), Fr17 (*p* ≤ 0.0001), and 3236 (*p* ≤ 0.0001) showed an increase in *ICAM1* expression, while 1138 (*p* ≤ 0.001), Fr9 (*p* ≤ 0.001), and ItVR (*p* ≤ 0.001) showed a significant decrease in expression compared to the control at this time point. SF2 was the only variant that showed an increase in *ICAM1* expression at all three time points studied, and ItVR showed a decrease in expression at the three time points studied. At the protein level, we observed a decrease in ICAM1 intensity 24 h post-transfection of the NA7 and ItVR variants, which aligns with the RNA expression data. At 48 h, the variants that showed an increase in band intensity corresponding to the RNA data were FR8 and 3236. However, the variant Fr17, which showed a remarkable increase in the expression of ICAM1 after transfection at the protein level, showed a decrease in band intensity at this time point. In addition, at 72 h post-transfection, we observed a decrease in the band intensity for the ICAM1 protein, with all of the nef variants aligning with the RNA data for the 118, Fr9, and ItVR variants, but not for the NA7, FR17, and 3236, which showed an increase in RNA expression instead (**Figure 3B**).

Similarly, *VCAM1* was also affected by the transfection of the Nef variants (**Figure 4**). At 24 h after the transfection of the NA7 variant, *VCAM1* expression significantly increased (*p* ≤ 0.001). In addition, *VCAM1* expression also significantly increased 48 h after the transfection of NA7 (*p* ≤ 0.0001), SF2 (*p* ≤ 0.0001), 1138 (*p* ≤ 0.0001), and Fr17 (*p* ≤ 0.0001). Interestingly, we observed a significant increase in *VCAM1* expression 72 h post-transfection with the 2044 variant (*p* ≤ 0.05).

At 24 h, there was a significant increase in expression of *SELE* gene in three Nef variants, SF2 (*p* ≤ 0.0001), NA7 (*p* ≤ 0.0001), and 3236 (*p* < 0.001) (**Figure 5**). At 48 h, there was an increase seen in NA7 (*p* ≤ 0.0001) and Fr17 (*p* < 0.0001). There was an upregulation in *SELE* at 72 h in PH Nef samples Fr9 (*p* ≤0.01) and Fr17 (*p* ≤ 0.0001), as well as in the non-PH variant NA7 (*p* ≤ 0.01).

### 3.4. Expression of Pro-Apoptotic BAX and BAK Genes in Nef-Treated Cells

It is generally well-accepted that pulmonary vascular cells undergo initial apoptosis as part of their response to vascular injury. In this experiment, we measured the expression of the *BAX* and *BAK* genes, which encode the Bcl-2 family proteins BAX and BAK, key regulators of apoptosis, in Nef-transfected pulmonary vascular cells. The expression of the *BAX* and *BAK* genes were measured using quantitative PCR at 24, 48, and 72 h post-Nef transfections. At 24 h, the transfection of most variants showed no significant increase in *BAX* expression, except for the Fr8 variant (*p* ≤ 0.001) (**Figure 6**). Interestingly, at 48 h post-transfection, *BAX* expression increased with SF2 (*p* ≤ 0.0001), as well as with the two PH-variants Fr17 (*p* ≤ 0.0001) and 3236 (*p* ≤ 0.0001), but decreased with the PH-variant ItVR (*p* ≤ 0.01). However, at 72 h, we only observed an increase in *BAX* expression following transfection with the 3236 variant (*p* ≤ 0.05) and a decrease following transfection with the 1138 (*p* ≤ 0.05) and Fr9 variants (*p* ≤ 0.05).

Similarly to what was observed in *BAX,* our results show that the most significant changes in *BAK* gene expression upon Nef expression in pulmonary vascular cells occurred 48 h post-transfection (**Figure 7**). *BAK* expression significantly increased upon 24 h of transfection of the NA7 (*p* ≤ 0.05), Fr8 (*p* ≤ 0.05), Fr9 (*p* ≤ 0.05), and 3236 (*p* ≤ 0.01) variants. *BAK* expression reached its highest level after 48 h with the NA7 (*p* ≤ 0.0001) variant. Additionally, it also significantly increased in SF2 (*p* ≤ 0.0001), 1138 (*p* ≤ 0.0001), and the two PH Nef variants, Fr17 (*p* ≤ 0.0001) and 3236 (*p* ≤ 0.01), compared to the control. At 72 h, *BAK* levels increased after the transfection of SF2 (*p* ≤ 0.05), 1138 (*p* ≤ 0.01), and Fr17 (*p* ≤ 0.01) variants.

### 3.5. Expression of Vasoconstrictive Endothelin-1 Gene in Nef-Transfected Pulmonary Vascular Cells

Endothelin-1 contributes to both the initial vasoconstriction response and subsequent vascular remodeling and repair. In this experiment, we sought to measure the gene expression of *EDN1* in Nef-transfected cells over time. Our results show significant increases in *EDN1* expression at 24 h post-transfection with the 2044 (*p* ≤ 0.001) and Fr8 (*p* ≤ 0.01) variants. At 48 h, there was a statistically significant increase in *EDN1* in additional variants NA7 (*p* ≤ 0.01) and SF2 (*p* ≤ 0.0001), as well as in PH variants Fr17 (*p* ≤ 0.0001) and 3236 (*p* ≤ 0.0001). At 72 h post-transfection, we observed an increase in *EDN1* expression with the SF2 (*p* ≤ 0.05) and 3236 (*p* ≤ 0.05) variants, while there was a decrease shown with the 1138 (*p* ≤ 0.05), ItVR (*p* ≤ 0.01), and FR17 (*p* ≤ 0.01) variants. Additionally, the transfection of the Fr9 variant did not affect *EDN1* gene expression at any of the tested time points. In contrast, the transfection of the SF2 variant increased the expression of *EDN1* at all three time points studied (**Figure 8A**). Furthermore, we studied the effect of the transient transfection of these variants at the protein level. At 24 h, we observed an increase in EDN1 with the FR8 variant, aligning with the RNA expression, but not with the 2044 variant. Additionally, we noted a decrease in EDN1 at the protein level with the 1138 variant, even though there were no significant changes at the RNA level. Additionally, at both 24 and 72 h, we observed a decrease in EDN1 across all of the variants studied, except for the 1138 variant at 24 h. The data at both time points did not align with the RNA results, except for the 1138 variant, where we observed a decrease in both RNA and protein levels at 72 h (**Figure 8B**).

### 3.6. Characterization of Inflammatory Cytokine Release Patterns in Nef-Treated Pulmonary Vascular Cells

Inflammation is a key component in endothelial dysfunction. The production of pro-inflammatory cytokines can further exacerbate endothelial dysfunction and contribute to vascular remodeling and disease progression. Little is understood about the inflammatory processes in Nef-mediated endothelial dysfunction, particularly in Nef isolates from HIV+ pulmonary hypertensive or normotensive individuals. Thus, we sought to profile the production of pro-inflammatory cytokines in Nef-treated pulmonary vascular cells in samples collected at 24, 48, and 72 h post-transfection. In this experiment, we measured IFNγ, IL-1β, IL-2, IL-4, IL-10, IL-12p70, IL-13, and TNFα. The BL1 control sample was used as a reference baseline for the interpretation of pro-inflammatory cytokine levels in pulmonary vascular endothelial and smooth muscle cell cocultures at different time points, summarized in **Figure 9**.

Most of the significant changes in pro-inflammatory cytokines were significant after 48 h in culture, particularly for IL-2, IL-4, IL-10, IL12p70, and IFNγ. Of the samples derived from HIV+ pulmonary normotensive controls, 2044 featured statistically significant decreases in the release of IL-2, Il-12p70, IFNg, IL-10, IL-1b, TNFa, and IL-13, mostly after 24 h. Both NA7 and SF2 exhibited significant increases in Th1 cytokines IL-2, Il-12p70, IFNg, and anti-inflammatory IL-10, as well as Th2 cytokine IL-13. The Nef constructs derived from HIV-pulmonary hypertensive donors demonstrated significant increases in Th1 cytokines IL-2, IL-12p70, and IFNg (**Figure 9A–C**), as well as anti-inflammatory IL-10 (**Figure 9D**), but did not elicit innate immunity cytokines IL-1b or TNFa (**Figure 9E,F**). Th2 cytokines IL-4 and IL-13 were significantly released from cells treated with PH-Nef, particularly after 48 h (**Figure 9G,H**).

### 3.7. HIV Nef Induces a Transient Increase in Endothelial Nitric Oxide Synthase (eNOS) Production in Pulmonary Vascular Cells In Vitro

Endothelial cells produce low levels of eNOS constitutively to help keep homeostatic vascular tone. However, factors like shear stress, increased insulin, estrogen, VEGF, and treatments with statins are known to stimulate eNOS production and/or secretion to reduce platelet aggregation and inflammation and promote angiogenesis, tissue repair, and overall vasodilation. In this experiment, the expression of eNOS was quantified in Nef-transfected pulmonary vascular endothelial and smooth muscle cell co-cultures supernatants at 24, 48, and 72 h post-transfection. The results show that there was an overall increase in eNOS in all samples (**Figure 10**).

When analyzed by sample, all Nef-transfected samples showed a statistically significant decrease in eNOS after 24 h, showing the time-dependent effect of eNOS expression (**Figure 10**). The expression of eNOS in the BL1 control did not change at any point in time, indicating its stability.

## 4. Discussion

This study describes the differential effects of HIV Nef variants in pulmonary vascular cells, with implications for endothelial cell activation, dysfunction, and inflammation. Our experimental design included Nef variants isolated from normotensive and pulmonary hypertensive HIV+ donors, which served to demonstrate significant differences in the expression of cell adhesion molecule genes *ICAM1*, *VCAM1*, *SELE*, pro-inflammatory cytokines, and nitric oxide production across multiple time points tested.

First, endothelial activation is an early step in vascular injury in which the expression of cell adhesion molecules such as ICAM1, VCAM1, E-selectin, P-selectin, and PECAM-1 facilitate the adhesion of leukocytes to the endothelium and the recruitment of inflammatory cells to the sites of injury. While we anticipated that Nef expression would enhance endothelial activation markers [34,35], our findings demonstrate this only to be true for SF2, NA7, Fr17, and 3236. Variant 1138 actually decreased *ICAM1* gene expression, but increased *VCAM1* expression, particularly at 48 h, and showed no significant changes in SELE expression, suggesting that this Nef variant may impair the first vascular activation steps in the response to vascular injury. Similarly, ITVR displayed a consistent decrease in *ICAM1* and no changes in *SELE* and *VCAM1* expression. Fr9 also presented a decrease in *ICAM1* and, in addition to this, had an increase in SELE expression. The 2044 sample displayed a significant increase in *VCAM1* expression, particularly at 72 h post-transfection, but no significant changes in *ICAM1* or *SELE* expression. Fr8 had no significant differences in *SELE* or *VCAM1* expression, but had an increase in *ICAM1* expression at 48 h. These findings suggest delayed or impaired vascular activation in variants with the inconsistent expression of endothelial cell activation markers.

Vascular remodeling and vasoconstriction are part of the second phase in the process of coping with vascular injury. Several vascular structural changes may occur at this stage. At this stage, apoptosis signaling in response to inflammation, oxidative stress, and vascular injury are common. Analyses of expression of apoptotic genes BAX and *BAK* demonstrated significantly increased apoptosis at one or more time point for NA7, SF2, 1138, Fr8, Fr9, Fr17, and 3236 variants. This increase in BAX or BAK expression at 48 h, but not at 24 h for several of the variants, may imply prolonged vascular cell stress and aberrant cell survival with potential implications in vascular remodeling. Interestingly, Fr9 and 1138 both had a statistically significant decrease in BAX expression at 72 h. In terms of vasoconstrictive signaling, the expression of endothelin-1 (*EDN1*) had a statistically significant increase at the early (24 h) time point only in the SF2, 2044, and Fr8 samples. Nef variants NA7, SF2, Fr17, and 3236 all had a statistically significant increase in *EDN1* expression at 48 h, but not at 24 or 72 h.

Third, the secretion of pro-inflammatory cytokines perpetuates a cascade of inflammatory events driving sustained inflammation within the pulmonary vascular microenvironment and ultimate disease progression. Our inflammatory cytokine analysis revealed that Nef variants derived from HIV-PH patients trended towards sustained increases in pro-inflammatory cytokines IL-2, IL-4, IL-13, IFNγ, and TNFα, which are critical mediators of vascular inflammation. The observations of increased IL-10 in cells treated with 3236 and Fr9, although not statistically significant, may suggest compensatory anti-inflammatory mechanisms that are insufficient to counterbalance the pro-inflammatory milieu.

Lastly, the vascular system engages in compensatory mechanisms such as vasodilation and the activation of anti-inflammatory signaling to maintain vascular tone and control inflammation in the restoration of vascular homeostasis. While the literature has shown that Nef decreases eNOS in porcine endothelial cells [36] and in coronary artery endothelial cells [35], our results demonstrate a transient increase at 24 h compared to the control. It is important to emphasize that the use of direct pulmonary vascular cell cocultures in our study may reveal compensatory mechanisms derived from cell–cell crosstalks to help maintain homeostasis. A potential interpretation is that the lack of sustained eNOS production in Nef-exposed cells may impair nitric oxide availability over time.

Our study is the first to characterize the impact of HIV Nef variants recovered from pulmonary hypertensive and normotensive HIV+ donors on pulmonary vascular cells in vitro. We acknowledge the limitations of our experimental approach, including the lack of physiological complexity contributed by shear stress and immune interactions of the pulmonary vasculature, the intrinsic variability of transient transfections, and the lack of direct clinical correlation that is lost when using cloned viral products. Nevertheless, the findings communicated in this study provide compelling evidence of the distinct effects of HIV Nef variants on endothelial cell activation, vasoconstrictive signaling, apoptosis, and inflammation. The differential responses observed across normotensive and PH-derived variants highlight the complex role of Nef in vascular dysfunction. Further understanding of how Nef variants interact with host genetic and environmental factors remains needed for developing comprehensive strategies to prevent and treat the vascular complications of viral infections.

## Figures and Tables

**Figure 1 idr-17-00065-f001:**
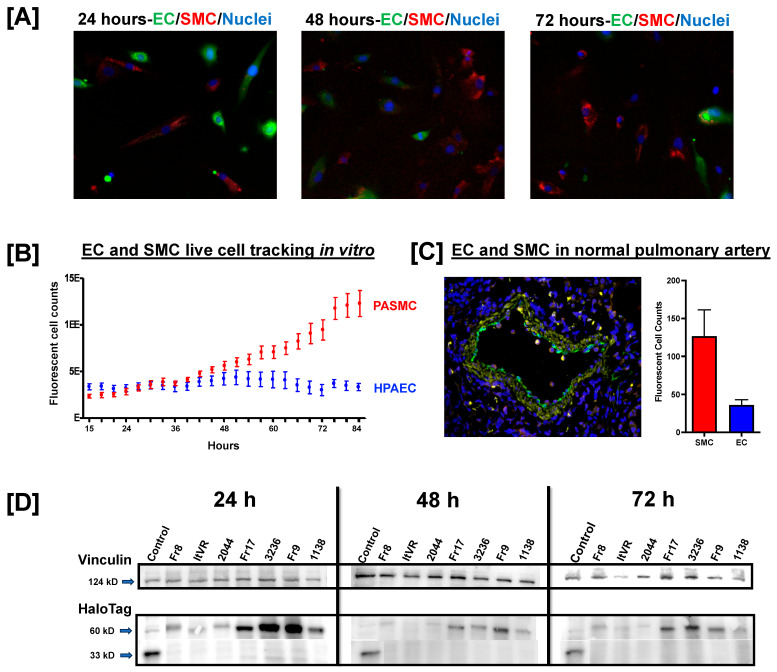
Endothelial and smooth muscle cell tracking in pulmonary vascular cell co-cultures, and confirmation of HIV Nef expression post-transfection. (**A**) Primary Human Pulmonary Artery Endothelial Cells (HPAEC) and Pulmonary Artery Smooth Muscle Cells (PASMC) were labeled with green and red cell-permeant dyes, respectively, seeded in direct cocultures, and tracked using live cell imaging in CELLCYTE X (Cytena). The vascular co-cultures contained HPAEC:PASMC seeded at a 1.8:1 ratio. Cell nuclei were stained with Hoechst prior to imaging at the indicated time points. (**B**) Quantification of green (endothelial) and red (smooth muscle cells) fluorescent objects over time. The smooth muscle cell population reached a three-fold excess over endothelial cells by the end of the 72+ hours assay. (**C**) Representative image of norma pulmonary artery stained for von-Willebrand factor (green) and smooth muscle actin (yellow) by immunofluorescence. Nuclei are shown in blue (DAPI). Fluorescent cell counts were documented using Image J and are shown in a bar graph. (**D**) Confirmation of HIV Nef expression by Nef-HaloTag immunoblotting. Pulmonary artery endothelial and smooth muscle cells were co-cultured and transfected with the indicated HIV Nef fusion constructs. Whole cell lysates from the transfected cells were immunoblotted with anti-HaloTag monoclonal antibody, followed by stripping and reprobing with human vinculin antibody as loading control.

**Figure 2 idr-17-00065-f002:**
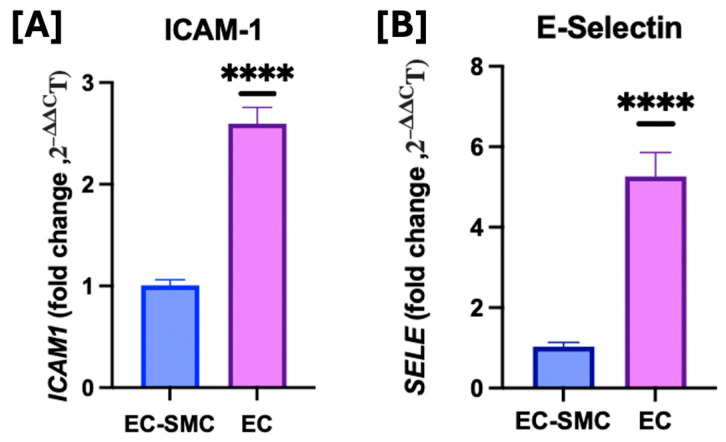
Effects of co-culture model on *ICAM1* and *SELE* gene expression. (**A**) *ICAM1* and (**B**) *SELE* mRNA transcripts were quantified using quantitative RT-PCR and normalized to RPLP0. An unpaired *t*-test was performed to determine statistical significance. Statistical significance: **** *p* < 0.0001.

**Figure 3 idr-17-00065-f003:**
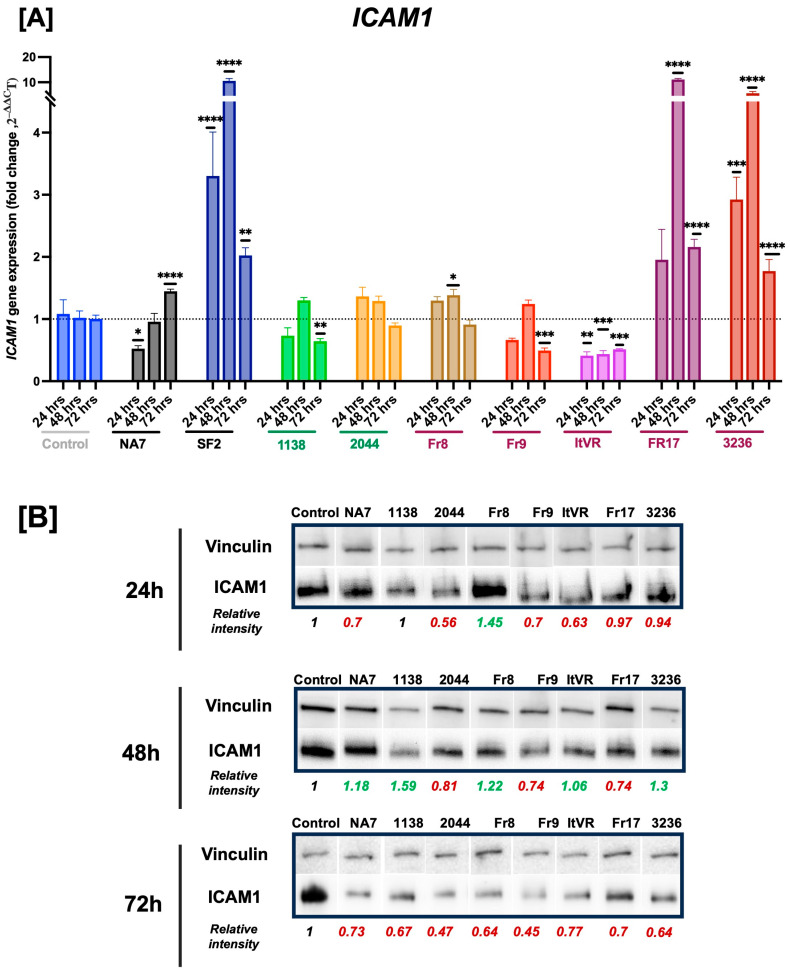
HIV Nef-induced changes in *ICAM1* gene expression. *ICAM1* mRNA transcripts were quantified using quantitative RT-PCR and normalized to RPLP0. Panel (**A**) shows the data from each Nef molecular clone plotted as a function of time (24, 48, and 72 h post Nef transfections). HIV Nef constructs from normotensive donors are labeled in green font; Nef constructs from pulmonary hypertensive donors are labeled in red font. Data are presented as mean ± SEM of quadruplicates. Statistical significance: * *p* < 0.05; ** *p* < 0.01; *** *p* < 0.001; **** *p* < 0.0001 compared to the empty vector BL1 control at every time point. Panel (**B**) shows immunoblots for ICAM1 with quantifications relative to housekeeping protein vinculin.

**Figure 4 idr-17-00065-f004:**
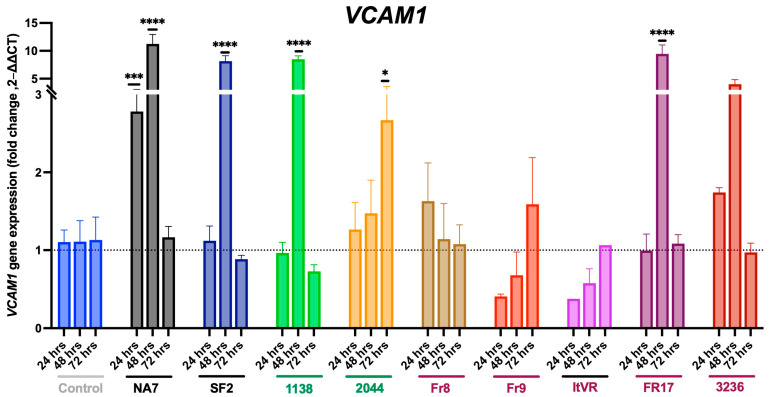
Nef-induced changes in VCAM1 gene expression. *VCAM1* mRNA transcripts were quantified by quantitative RT-PCR and normalized to RPLP0. HIV Nef constructs from normotensive donors are labeled in green font; Nef constructs from pulmonary hypertensive donors are labeled in red font. Data was plotted by molecular construct, each time point normalized to its respective control. Data are presented as mean ± SEM of quadruplicates. Statistical significance: * *p* < 0.05; *** *p* < 0.001; **** *p* < 0.0001 compared to the empty vector BL1 control at every time point.

**Figure 5 idr-17-00065-f005:**
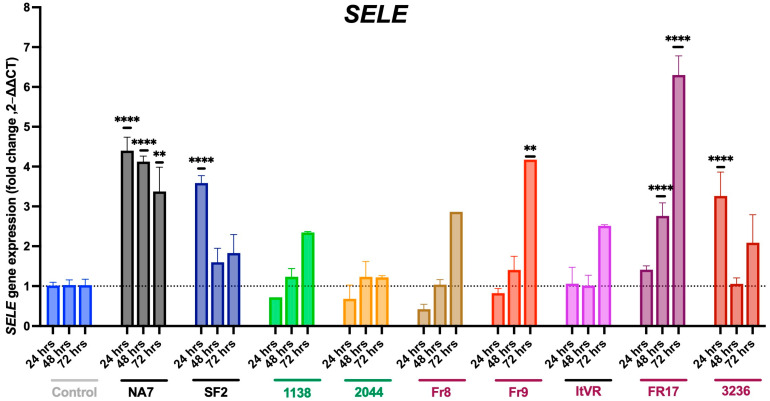
Nef-induced changes in *SELE* gene expression. *SELE* mRNA transcripts were quantified by quantitative RT-PCR and normalized to RPLP0. HIV Nef constructs from normotensive donors are labeled in green font; Nef constructs from pulmonary hypertensive donors are labeled in red font. Data was plotted by molecular construct, each time point normalized to its respective control. Data are presented as mean ± SEM of quadruplicates. Statistical significance: ** *p* < 0.01; **** *p* < 0.0001 compared to the empty vector BL1 control at every time point.

**Figure 6 idr-17-00065-f006:**
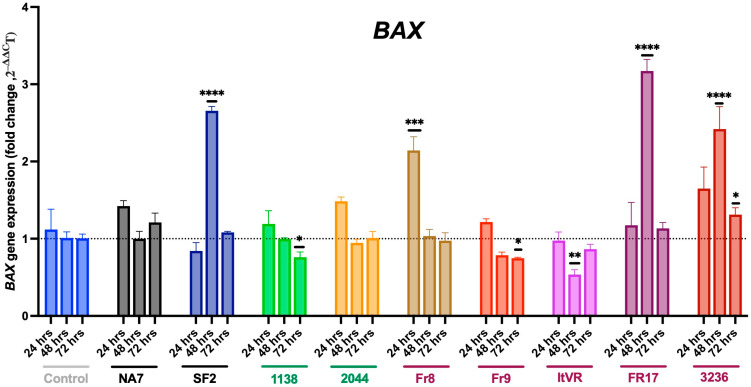
Delayed or impaired pro-apoptotic BAX signaling in Nef-transfected pulmonary vascular cells. *BAX* mRNA transcripts were quantified using quantitative RT-PCR and normalized to *RPLP0.* HIV Nef constructs from normotensive donors are labeled in green font; Nef constructs from pulmonary hypertensive donors are labeled in red font. Data are presented as mean ± SEM of quadruplicates. Statistical significance: * *p* < 0.05; ** *p* < 0.01; *** *p* < 0.001; **** *p* < 0.0001 compared to the empty vector BL1 control at every time point.

**Figure 7 idr-17-00065-f007:**
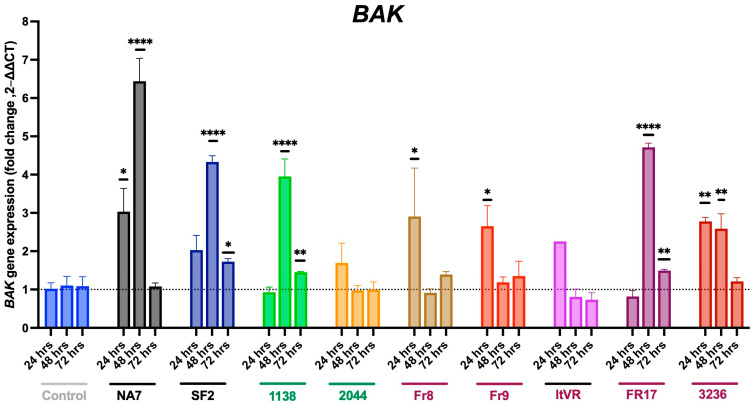
Delayed or impaired pro-apoptotic BAK signaling in Nef-transfected pulmonary vascular cells. *BAK* mRNA transcripts were quantified using quantitative RT-PCR and normalized to *RPLP0.* HIV Nef constructs from normotensive donors are labeled in green font; Nef constructs from pulmonary hypertensive donors are labeled in red font. Data are presented as mean ± SEM of quadruplicates. Statistical significance: * *p* < 0.05; ** *p* < 0.01; **** *p* < 0.0001 compared to the empty vector BL1 control at every time point.

**Figure 8 idr-17-00065-f008:**
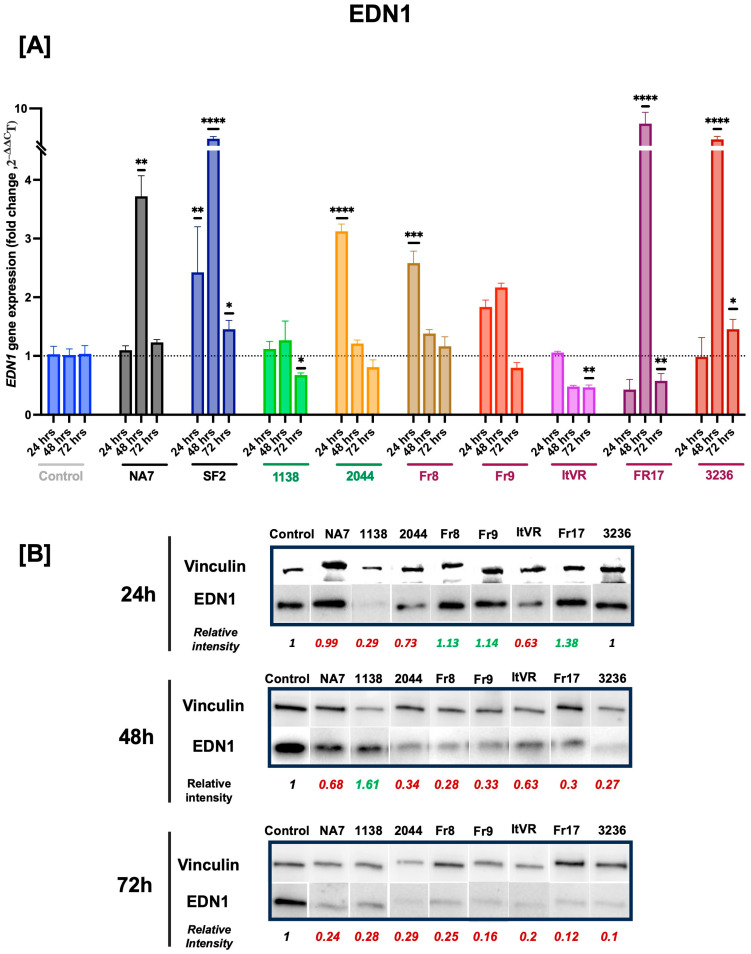
HIV Nef induces a transient increase in endothelin-1 gene expression in pulmonary vascular cells. *EDN1* mRNA transcripts were quantified using quantitative RT-PCR and normalized to RPLP0. Panel (**A**) shows the data from each Nef molecular clone plotted as a function of time (24, 48, and 72 h post Nef transfections). HIV Nef constructs from normotensive donors are labeled in green font; Nef constructs from pulmonary hypertensive donors are labeled in red font. Data are presented as mean ± SEM of triplicates or quadruplicates. Statistical significance: * *p* < 0.05; ** *p* < 0.01; *** *p* < 0.001; **** *p* < 0.0001 compared to the empty vector BL1 control at every time point. Panel (**B**) shows immunoblots for endothelin-1 with quantifications relative to housekeeping protein vinculin at every time point.

**Figure 9 idr-17-00065-f009:**
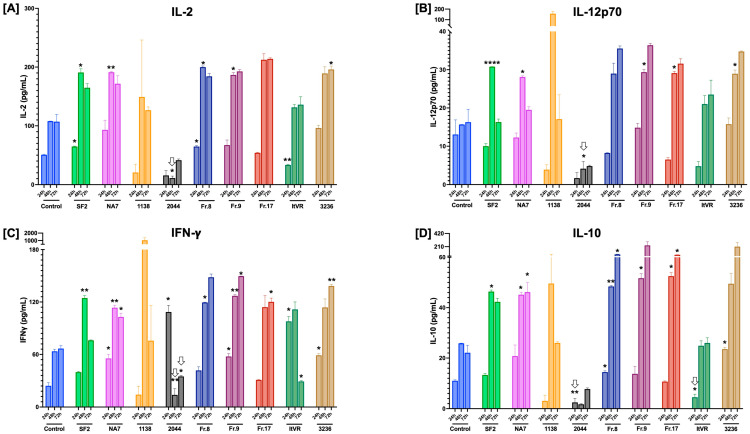
Impact of HIV Nef variants on inflammatory cytokine secretion in pulmonary vascular cells in vitro. Pulmonary arterial endothelial and smooth muscle cells were co-cultured, transfected with HIV Nef molecular constructs (or control vector), and incubated for 24, 48, or 72 h post-transfection. Cell culture supernatants were collected and analyzed for pro-inflammatory cytokine levels using Meso Scale Discovery electrochemiluminescence technology. The bar graphs show pg/mL concentrations of each detected cytokine (IL-2 in Panel (**A**), IL-12p70 in Panel (**B**), IFNg in Panel (**C**), and IL-10 in Panel (**D**), IL-1b in Panel (**E**), TNFa in Panel (**F**), IL-4 in Panel (**G**), and IL-13 in Panel (**H**); all labeled for each chart) at three time points (24, 48, or 72 h) in cells transfected with each Nef molecular construct (experimental groups labeled in the X-axis). Data are clustered by Nef construct. Statistical analyses were performed using one-way ANOVA by time point. All statistically significant differences shown for each sample refer to increased cytokine release compared to their respective control at 24, 48, or 72 h. Significant decreases are denoted by down arrows. Statistical significance: * *p* < 0.05; ** *p* < 0.01; **** *p* < 0.0001 compared to the empty vector BL1 control at each time point.

**Figure 10 idr-17-00065-f010:**
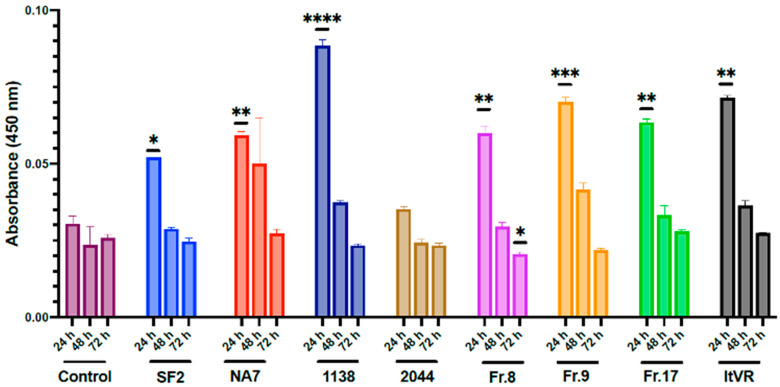
Nef-induced transient increase in endothelial nitric oxide synthase (eNOS) in pulmonary vascular cells in vitro. The release of eNOS was measured using ELISA at 24, 48, and 72 h post-transfection of pulmonary vascular cells with Nef molecular constructs. Data plotted with a molecular construct. Data are presented as mean ± SEM of triplicates. Statistical significance: * *p* < 0.05; ** *p* < 0.01; *** *p* < 0.001, **** *p* < 0.0001 compared to the empty vector BL1 control at every time point.

**Table 1 idr-17-00065-t001:** List of primers used for quantitative real-time PCR. The primer pairs listed below were designed in-house using NCBI Primer-BLAST and were synthesized by Integrated DNA Technologies. Primer specificity was validated using a standard PCR to confirm amplification of single specific amplicons prior to use in qPCR assays.

Gene Symbol	Gene ID	Protein Name	Primer Sequence	Amplicon Size
*BAX*	581	BCL2 associated X, apoptosis regulator	Fwd: 5′-GGACGAACTGGACAGTAACA-3′ Rev: 5′-ACCACCCTGGTCTTGGAT-3′	271 bp
*BAK*	578	BCL2 antagonist/killer 1	Fwd: 5′-ACGCTATGACTCAGAGTTCC-3′ Rev: 5′-CTTCGTACCACAAACTGGCC-3′	360 bp
*EDN1*	1906	Endothelin-1	Fwd: 5′-AGAGTGTGTCTACTTCTGCC-3′ Rev: 5′-GTTGTGGGTCACATAACG-3′	442 bp
*ICAM1*	3383	Intercellular cell adhesion molecule-1	Fwd: 5′-AGCCAGTGGGCAAGAACCTT-3′ Rev: 5′-CGGCACGAGAAATTGGCTCC-3′	187 bp
*SELE*	6401	E-selectin	Fwd: 5′-GGCAGTGGACACAGCAAATC-3′ Rev: 5′-TGGACAGCATCGCATCTCA-3′	224 bp
*VCAM1*	7412	Vascular cell adhesion molecule-1	Fwd: 5′-TGGTCGTGATCCTTGGAGCC-3′ Rev: 5′-AGATGTGgTCCCCTCATTCGT-3′	221 bp

## Data Availability

The data generated or analyzed during this study are included in the article. Further details will be made available upon request.

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
