# Peer review of "Differential Effects of Human Immunodeficiency Virus Nef Variants on Pulmonary Vascular Endothelial Cell Dysfunction"

_2036-7449, 2025, doi:10.3390/idr17030065_

Round 1

Reviewer 1 Report

Comments and Suggestions for Authors

Differential Effects of Human Immunodeficiency Virus Nef Variants on Pulmonary Vascular Endothelial Cell Dysfunction

Pulmonary hypertension (PH) is a common and severe comorbidity observed in patients living with HIV. Expression of the HIV protein, Nef persists in pulmonary hypertension patient lungs. The group showed previously that Nef polymorphisms were identified in HIV positive individuals with PH. Therefore, the authors utilise the transient transfection of pulmonary endothelial and smooth muscle cells with Nef variants known to be associated with HIV-positive pulmonary hypertension to investigate known markers of endothelial cell dysfunction. The number of Nef variants investigated is comprehensive as they interrogate both HIV+PAH and normotensive constructs. They also utilise several techniques to investigate endothelial dysfunction. Unfortunately, there doesn’t appear to be a clear correlation between all PAH Nef variants and vascular dysfunction, as indicated by the title of study. In fact, there isn’t a clear correlation between any of the Nef variants. This indeed could be biological, but I think the authors need to consider the impact of transient transfection and the use of endothelial cells and smooth muscle cells in combination.

 Major comments

Comment 1: The use of the different Nef variants associated with disease is a strength of the study but how did the authors account for the possibility that there would be variation in transfection efficiency? Was the expression equivalent across the different constructs? This could impact on the various vascular dysfunction endpoints and the authors need to demonstrate that the expression of the Nef variants was either equivalent or the data is normalised to transfection expression.

Comment 2: I have some concerns/suggestions regarding the co-culture experiments, which underpin this study. Firstly, it isn’t clear where the 1.8:1 ratio of endothelial and smooth muscle cells comes from. Why was this chosen? Did the authors make sure that the ratio of endothelial cells and smooth muscle cells were maintained during the timecourse?

I have two suggestions. 1) I would like to see the endothelial cells and smooth muscle cells labelled in culture across the timecourse, does the ratio survive the treatment timecourse following transfection? 2) Relating to comment 1, was the transfection efficiency equivalent between the endothelial cells and smooth muscle cells? 3) I would like to see (some of) these experiments with endothelial cells and smooth muscle cells alone. This will allow a better understanding of the contribution of mono- and co-culture in experiments that predominantly focus on endothelial cell dysfunction.

Comment 3: Although the authors have tried to assess multiple aspects of vascular dysfunction focussing on genetic changes, cytokines and nitric oxide I find the results difficult to interpret. The only dataset that is understandable is the eNOS dataset where all Nefs (apart from 2044) elevate eNOS at 24 hours compared to control. The authors discuss that reduced eNOS over the time course aligns with previous studies. This doesn’t align with previous literature as the levels are the same as the control, Nef reduced eNOS versus control in the coronary artery paper cited.

As the authors correctly suggest they expected ICAM1 gene expression to be elevated, but only NA7 induces this. As the authors discuss there are other cell adhesion markers modified in vascular dysfunction. Therefore, what happens to E-selectin, VCAM etc.? This would strengthen the claims of the authors regarding the effects of the Nefs from HIV+ patients, and the downregulation of cell adhesion molecules.

Similar with Bad there are other markers of apoptosis associated with vascular dysfunction, what about Bak? Again, this would give the reader more confidence that the differential changes between the Nef variants are not specific to one gene. Alternatively, what effect do the Nef variants have upon apoptosis?

Finally, the EDN dataset is problematic as there are only 2 replicates for some of the datapoints including the BL1 control, therefore how has statisitical analysis been conducted on n=2? Also, the data doesn’t match between the timepoint data and construct data. See NA7 at 24 and 48 hours.

The MSD panel needs a complete overhaul as the authors discuss trends and sustained expression when there isn’t any statistical significance. They also say that the FR1 control is similar to the BL1 control validating consistency but there are statistically significant differences between these controls. I am afraid the stats don’t make any sense to this reviewer. For example, why is IL13 levels in 2044 not statistically different from BL1 at 24 hours? There are too many examples to list that are similar to this. Therefore this data does not support any conclusions in the discussion.

Why does the control increase NO production at 48 hours? Also, how many times was this experiment conducted? The figure legend says duplicate. Is this two experiments? Please conduct more if this is the case.

Comment 4: Overall, the gene expression studies are difficult to understand. Looking at the ICAM1 and BAX data I am confused by the variability in the BL1 24-, 48- and 72-hour samples. Surely, this is the reference sample at each timepoint and should be a fold change of 1. Could this explain the variability?

Comment 5: The manuscript is well written but needs improvement in key areas. 1) The citations are often to manuscripts which don’t directly describe the statement in the text. For example, (page 8 line 238) “Several forms of PH including PAH, CTEPH, PH associated with lung disease or due to left heart disease are associated with increased levels of ET-1[32].” In this example I could not find any mention of endothelin or PH in this reference. Also, “Inflammation is a key component in endothelial dysfunction [33], this statement is connected to a citation in the manuscript referenced.  Please correct all incidences of this. 2) To help with this, please put references in the opening paragraph of the results section (page 5 line 174-180). This reference would be a good starting point – PMID: 30220106. 3) Similarly, a reference to corroborate the claim regarding Bax and vascular dysfunction would be helpful. These suggestions also apply to endothelin-1, inflammation markers, eNOS and NO production.

Comment 6: The authors describe the variants in the intro as polymorphisms, but then describe them as construct Ids in the methods and results. A table describing the polymorphisms, constructs and disease would be helpful to then follow the subsequent results of HIV+ PAH versus normotensive.

Author Response

Comment 1: The use of the different Nef variants associated with disease is a strength of the study but how did the authors account for the possibility that there would be variation in transfection efficiency? Was the expression equivalent across the different constructs? This could impact on the various vascular dysfunction endpoints and the authors need to demonstrate that the expression of the Nef variants was either equivalent or the data is normalised to transfection expression.

Response: We appreciate your comment regarding the use of different Nef variants in our study. You are correct, there are intrinsic variations in transfection efficiency. Our Nef clones have the same HaloTag parental vector, we utilized the same expansion, purification, and transfection systems across the board. However, it is not unusual to observe variable expression levels across Nef panels, probably due to intrinsic properties of the encoded Nef protein. While the reasons for variations in Nef transfection efficiencies are beyond the scope of this work, we may speculate that several factors influencing Nef protein expression may be at play when examining Nef mutants including different interactions with chromatin or nuclear localization, different abilities to affect premature protein degradation by ubiquitination, increased susceptibility of some Nef to misfolding, aggregation, or differences at the level of mRNA stability leading to potential degradation of transcripts at different levels.

In this resubmission, we include evidence of Nef-HaloTag protein expression by Western blot (Figure 1, panel D). With these data, we confirm transfection efficiency in all the samples tested and moreover, it shows that some Nef variants have higher expression levels than others due to their intrinsic differences. While normalized protein datasets are fairly standard in research, we feel that normalizing the results to Nef expression would not accurately represent the natural activity of Nef.

Comment 2: I have some concerns/suggestions regarding the co-culture experiments, which underpin this study. Firstly, it isn’t clear where the 1.8:1 ratio of endothelial and smooth muscle cells comes from. Why was this chosen? Did the authors make sure that the ratio of endothelial cells and smooth muscle cells were maintained during the timecourse?

I have two suggestions. 1) I would like to see the endothelial cells and smooth muscle cells labelled in culture across the timecourse, does the ratio survive the treatment timecourse following transfection? 2) Relating to comment 1, was the transfection efficiency equivalent between the endothelial cells and smooth muscle cells? 3) I would like to see (some of) these experiments with endothelial cells and smooth muscle cells alone. This will allow a better understanding of the contribution of mono- and co-culture in experiments that predominantly focus on endothelial cell dysfunction.

Response: Thank you for this important question and suggestions. In our experience, primary PASMC have a faster growth rate HPAEC. Thus, when co-cultures are established with EC and SMC seeded 1:1, the PASMC outgrows the EC in a non-physiological manner. Hence, we have implemented laboratory protocols so that we seed the HPAEC and PASMC at 1.8:1 to allow proper representation on EC in our experiments. Following your suggestion, we labeled EC and SMC with green and red-fluorescent dyes, respectively, to track their fate in culture. Our results show that at the end of the experiments, the proportion of cells ends at 3 SMC per EC (Figure 1 in the revised manuscript). To confirm physiological relevance, we quantified the SMC and EC in normal pulmonary arteries in formalin-fixed paraffin-embedded lungs (n=3) that were double-stained for vWF and alpha-smooth muscle actin. The results of the quantification showed that there are 3.42 SMC per EC, which supports the physiological relevance of our cell culture approach in terms of cell numbers. The revised manuscript now includes this critical methodological aspect in Figure 1.

Regarding your requested experiments with endothelial cells alone and in coculture to examine the contribution of the cell populations to our experiments, we examined the effect of the co-culture model on baseline expression of ICAM1 and SELE in endothelial cell monocultures compared to endothelial and smooth muscle cell co-cultures. We found that when compared to EC-SMC co-cultures, EC only monocultures had a statistically significant increase in baseline ICAM1 and E-selectin expression (p=<0.0001) (Figure 2). Our results align with those in the literature, the expression of cell adhesion molecules in EC monocultures compared to EC and SMC co-cultures and found cell adhesion molecules to be elevated in EC monocultures (Fan, et al; doi:10.1152/ajpheart.01029.2009). Since our results align with those in the literature, this work focused on the impact of HIV Nef on pulmonary cell biology using direct coculture systems.

Comment 3: Although the authors have tried to assess multiple aspects of vascular dysfunction focussing on genetic changes, cytokines and nitric oxide I find the results difficult to interpret. The only dataset that is understandable is the eNOS dataset where all Nefs (apart from 2044) elevate eNOS at 24 hours compared to control. The authors discuss that reduced eNOS over the time course aligns with previous studies. This doesn’t align with previous literature as the levels are the same as the control, Nef reduced eNOS versus control in the coronary artery paper cited.

Response: Great comment. With Nef being an enigmatic and multifaceted protein, the results of investigating multiple Nef variants in one study just gets more complicated and hard to compare with other studies in the literature. Regarding the transient increases in eNOS, thank you for noting the incorrect reference of published literature. While the literature has shown that Nef decreases eNOS in porcine endothelial cells and in coronary artery endothelial cells, our results demonstrate a transient increase at 24 hours, compared to control. It is important to emphasize that the use of direct pulmonary vascular cell cocultures in our studies may reveal compensatory mechanism derived from cell-cell crosstalk to help maintain homeostasis. A potential interpretation is that the lack of sustained eNOS production in Nef-exposed cells may impair nitric oxide availability over time. We have revised the Discussion section in this resubmission to reflect these points.

As the authors correctly suggest they expected ICAM1 gene expression to be elevated, but only NA7 induces this. As the authors discuss there are other cell adhesion markers modified in vascular dysfunction. Therefore, what happens to E-selectin, VCAM etc.? This would strengthen the claims of the authors regarding the effects of the Nefs from HIV+ patients, and the downregulation of cell adhesion molecules.

Response: We appreciate your inquiry regarding analysis of additional cell adhesion markers E-selectin and VCAM-1 to strengthen our argument that the Nef variant may impair the first vascular activation steps in the response to vascular injury. Your comment prompted us to further examine gene expression of VCAM1 and SELE.  Pulmonary hypertensive Nef variants showed, largely an impaired ability to express cell adhesion molecules, In addition to this, only one non-PH Nef showed an upregulation of VCAM1 expression at 24 hours. At 48 hours, two of the additional Nef variants, SF2 and NA7 as well as 1138, a normotensive variant. Of note, Fr17, a PH variant had an upregulation of VCAM1 expression. At 72 hours, none of the Nef variants displayed significant changes in VCAM-1 expression.

ICAM1 followed a similar trend, in that none of the PH variants displayed an upregulation of ICAM1 at 24 hours post-transfection. Similar to ICAM-1, at 48 hours there was a significant upregulation of ICAM1 at 48 hours, particularly in SF2, Fr17, and 3236. This was followed by an overall decrease in ICAM1 expression at 72 hours in all except NA7. This highlights the either delay or lack of elevated cell adhesion molecules. As for E-selectin, at 24 hours, there was a significant increase in expression of SELE gene in only two Nef variants, SF2 and NA7. At 48 hours, there was an increase seen in NA7, Fr17, and 1138. There was an upregulation in SELE at 72 hours in several PH Nef samples Fr8, Fr9, ItVR, and Fr.17, opposite of the trend to decrease at 72 hours that was seen in ICAM1 and VCAM1. These data are now included in this resubmission (new Figures 4-5).

Similar with Bad there are other markers of apoptosis associated with vascular dysfunction, what about Bak? Again, this would give the reader more confidence that the differential changes between the Nef variants are not specific to one gene. Alternatively, what effect do the Nef variants have upon apoptosis?

Response: Thank you for bringing to our attention that other markers associated with vascular dysfunction, such as BAK would strengthen our argument that the differential changes seen are not specific to one gene. Your comment prompted us to further examine gene expression of BAK. At 24 hours, only two Nef variants, NA7 and 3236, had a significant increase in BAK gene expression. At 48 hours, SF2, NA7, Fr17, 1138, and 3236 all had statistically significant increases in BAK expression, followed by an overall trend to decrease BAK expression at 72 hours. BAX expression was the highest at 48 hours in two PH Nef variants Fr17 and 3236 as well as in SF2. Similar to what was observed in BAK, BAX expression declined in these samples at 72 hours post-transfection. Overall, these results suggest that there was a delayed apoptotic response. The new data are now included in this resubmission (new Figure 7).

Finally, the EDN dataset is problematic as there are only 2 replicates for some of the datapoints including the BL1 control, therefore how has statisitical analysis been conducted on n=2? Also, the data doesn’t match between the timepoint data and construct data. See NA7 at 24 and 48 hours.

Response: We sincerely appreciate you bringing to our attention that several data points were missing from our figures. We agree that statistical analysis should be conducted on n=3, at minimum. Three separate transfections were conducted to support this manuscript. In addition, 4 technical replicates were included per each sample per each gene included in this manuscript. We have included re-analyzed qPCR data to support our findings.

After reviewing the original way our data was submitted, we agree that normalizing both to the 24-hour time point for each sample and to the BL1 control led to confusion. To resolve this, we decided to normalize to the BL1 control only, while still displaying all the time points for each variant together, so that the reader can visualize what changes are occurring at each time point. The new data are now included in this resubmission (new Figure 8).

The MSD panel needs a complete overhaul as the authors discuss trends and sustained expression when there isn’t any statistical significance. They also say that the FR1 control is similar to the BL1 control validating consistency but there are statistically significant differences between these controls. I am afraid the stats don’t make any sense to this reviewer. For example, why is IL13 levels in 2044 not statistically different from BL1 at 24 hours? There are too many examples to list that are similar to this. Therefore this data does not support any conclusions in the discussion.

Response: Thank you for your inquiry about our inflammatory cytokine datasets. We now realize that our attempt to simplify data presentation by using heatmaps made it more difficult and misled the statistical approach and ultimately, its interpretation. We have re-analyzed the datasets, and we are presenting the data now as bar graphs for each cytokine grouped by Nef variant, by timepoint. We performed the statistical analyses using one-way ANOVA by time-point, and the statistically significant differences are shown for each sample compared to their respective control at 24, 48, or 72 hours. We found that most of the significant changes in pro-inflammatory cytokines were significant after 48 hours in culture, particularly for IL-2, IL-4, IL-10, IL12p70, and IFNγ. Of the samples derived from HIV+ pulmonary normotensive controls, 2044Nef featured statistically sig-nificant decreases in the release of IL-2, Il-12p70, IFNg, IL-10, IL-1b, TNFa, and IL-13, mostly after 24 hours. Both NA7 and SF2Nef exhibited significant increases in Th1 cyto-kines IL-2, Il-12p70, IFNg and anti-inflammatory IL-10, as well as Th2 cytokine IL-13. The Nef constructs derived from HIV-pulmonary hypertensive donors demonstrated signifi-cant increases in Th1 cytokines IL-2, IL-12p70 and IFNg (Figure X, Panels A-C), as well as anti-inflammatory IL-10 (Figure X, Panel D), but did not elicit innate immunity cytokines IL-1b or TNFa (Figure X, Panels E-F). Th2 cytokines IL-4 and IL-13 were significantly released from cells treated with PH-Nef, particularly after 48 hours (Figure X, Panels G-H). The new data are now included in this resubmission (new Figure 9).

Why does the control increase NO production at 48 hours? Also, how many times was this experiment conducted? The figure legend says duplicate. Is this two experiments? Please conduct more if this is the case.

Response: Thank you for your comment. Yes, additional experiments are required, particularly to clarify unexpected results. Unfortunately, we couldn’t complete additional determinations of NO during this resubmission time period and have decided to remove this dataset from the manuscript.

Comment 4: Overall, the gene expression studies are difficult to understand. Looking at the ICAM1 and BAX data I am confused by the variability in the BL1 24-, 48- and 72-hour samples. Surely, this is the reference sample at each timepoint and should be a fold change of 1. Could this explain the variability?

Response: Correct, the expression of the target gene (ICAM1, VCAM1, EDN1, SELE, or EDN1) was normalized to the expression of the housekeeping gene (RPLP0) in each BL1 replicate at each time point to account for possible variation between replicates. These had slight variations, all near an average fold change of 1, at 24, 48, and 72 hours. There were completely different BL1 controls at each time point to ensure any changes in gene expression of the target gene was truly biological. Gene expression changes seen in all other samples were relative to the BL1 controls at each respective time point.

Comment 5: The manuscript is well written but needs improvement in key areas. 1) The citations are often to manuscripts which don’t directly describe the statement in the text. For example, (page 8 line 238) “Several forms of PH including PAH, CTEPH, PH associated with lung disease or due to left heart disease are associated with increased levels of ET-1[32].” In this example I could not find any mention of endothelin or PH in this reference. Also, “Inflammation is a key component in endothelial dysfunction [33], this statement is connected to a citation in the manuscript referenced.  Please correct all incidences of this. 2) To help with this, please put references in the opening paragraph of the results section (page 5 line 174-180). This reference would be a good starting point – PMID: 30220106. 3) Similarly, a reference to corroborate the claim regarding Bax and vascular dysfunction would be helpful. These suggestions also apply to endothelin-1, inflammation markers, eNOS and NO production.

Response: Thank you for the suggestions. We identified leftover parenthesis for citations that were not linked to the list of references. We took care of it in this resubmission.

Comment 6: The authors describe the variants in the intro as polymorphisms, but then describe them as construct Ids in the methods and results. A table describing the polymorphisms, constructs and disease would be helpful to then follow the subsequent results of HIV+ PAH versus normotensive.

Response: Thank you for the suggestion. We took care of it in this resubmission.

Reviewer 2 Report

Comments and Suggestions for Authors

Authors did interesting work to address HIV-related pulmonary complications by focusing on the role of Nef variants in pulmonary vascular dysfunction, as it is bridging the gap HIV pathophysiology. Authors used multiple endothelial dysfunction markers (ICAM1, BAX, EDN1, eNOS, cytokines) to describes the differential effects of HIV Nef variants in pulmonary vascular cells, with implications for endothelial cell activation, dysfunction, and inflammation.

Major Comment:

    1. Does authors also tried to check the protein level expression of any marker they have used for the study, as we know transcript levels not always reflect the translation, but that can strengthen their conclusion.

All the best!

Author Response

Authors did interesting work to address HIV-related pulmonary complications by focusing on the role of Nef variants in pulmonary vascular dysfunction, as it is bridging the gap HIV pathophysiology. Authors used multiple endothelial dysfunction markers (ICAM1, BAX, EDN1, eNOS, cytokines) to describes the differential effects of HIV Nef variants in pulmonary vascular cells, with implications for endothelial cell activation, dysfunction, and inflammation.

Major Comment:

    1. Does authors also tried to check the protein level expression of any marker they have used for the study, as we know transcript levels not always reflect the translation, but that can strengthen their conclusion.

Response: Thank you for your comment and request. We absolutely agree that post-transcriptional events influence protein translation and hence, biological significance of gene expression data. Our revised manuscript now shows Western blot data for ICAM1 and endothelin (Figures 3 and 8).

Round 2

Reviewer 1 Report

Comments and Suggestions for Authors

The authors have done a tremendous amount of work to address my comments, and I commend them on this. Now that I can see the expression for the constructs I can start to interpret the data better. I am afraid some of the conclusions are not matched by the data, particularly the adhesion and apoptosis qPCR. This needs to be fully interpretated as it feels to this reader that non-significant trends are being used to back the authors conclusions. Please can the authors revisit the qPCR data and make it clear whether they are discussing changes versus the BL1 control at each timepoint, or the changes in gene expression across the timecourse for each individual Nef construct, as I am still confused.

I have diligently been through the manuscript again and have written my interpretations based upon the data in the figures. Overall, eNOS is transiently up regulated by all Nefs, apart from 2044. All Nefs cause significant changes in pro-inflammatory cytokines, again 2044 appears to do the opposite, and all constructs reduce endothelin-1, although I would like to see multiple blots with all samples on the same membrane so densitometry could be conducted.

Comment 1: It is clear that there are discrepancies between the transfection expression. It appears that Fr17, 3236, Fr9 and 1138 are more highly expressed compared to Fr8, ItVR and 2044. There is no evidence of the NA7 and SF2 expression levels. I would suggest a limitations of study paragraph in the discussion with regards to the use of transient transfections. Ideally cells from individuals with these mutations would be used, which I appreciate are difficult to source.

Comment 2: Thank you for including VCAM1 and SELE to allow a fuller understanding of the effects of the constructs on EC/vascular dysfunction genes. My interpretation of the qPCR results compared to the BL1 control at the equivalent timepoint is as follows, and doesn’t match the interpretation for ICAM1 in the manuscript:

ICAM1 qPCR

Down

24 hrs – NA7, ItVR

48 hrs – ItVR

72 hrs – 1138, Fr9, ItVR

Up

24 hrs – SF2, FR17(?), 3236(?)

48 hrs – SF2, Fr8, FR17, 3236

72 hrs – NA7, SF2, Fr8, FR17, 3236

VCAM1 qPCR

Down

24 hrs – Fr9(?), ItVR(? Only 1 datapoint)

48 hrs –

72 hrs –

Up

24 hrs – NA7

48 hrs – SF2, 1138, FR17, 3236(?)

72 hrs – 2044(?)

SELE qPCR – control is variable at 24 hours.

Down

24 hrs –

48 hrs –

72 hrs –

Up

24 hrs – NA7, SF2, 3236

48 hrs – NA7, 1138, FR17

72 hrs – NA7, SF2(?), 1138(?), Fr8, Fr9, ItVR, FR17, 3236(?).

I would conclude that NA7, SF2, FR17, 3236 increase ICAM1, VCAM1 and SELE. Interestingly 1138 decreased ICAM1 but increases VCAM1 and SELE. Fr9 and ItVR appear to downregulate ICAM1 and VCAM1 but interestingly are up at 72 hours.

Comment 3: Having looked in more detail I am concerned how the authors have normalised the data to generate the delta delta CT values. If these are delta delta CT then BL1 controls should all be 1, which they aren’t. How did the authors generate the fold change?

Comment 4: There is no mention of the ICAM-1 western data in the results section, is this because it largely doesn’t agree with the qPCR data, as all variants decrease ICAM-1 expression at 72 hours?

Comment 5: Thank you for including another apoptotic gene, although the authors have neglected to tell the readers about BAK. Please add this. Again, your interpretation of the qPCR data isn’t quite accurate. Fr8 significantly increases BAX at 24 hours (and potentially BAK). Also the authors say that ItVR didn’t change over time but statistically BAX expression was decreased at 48 hours. I would argue that only SF2 and FR17 have the same (significant) profiles between the two apoptosis genes. The other observations are just trends, which could be rectified by using the correct delta delta CT calculation.

Comment 6: The data convincingly shows a reduction in the endothelin-1 protein, however, these don’t seem to be from the same blots. How many times was this data repeated?

Comment 7: Thank you for clarifying the cytokine data, this is more convincing and gives a clear pattern especially at the later timepoints. The authors might want to revisit the stats as looking at some of the individual data points, I believe they are missing some important conclusions. For example, IL2 levels in FR17 at 48 hours, surely this is significant?

Minor Comment 1: I wonder whether some of the methodology would be best in a supplementary section? For example, could the transfection efficiency be Supplementary Figure 1? And then the ICAM-1 and E-selectin qPCR could be incorporated into Figure 1, which will help with the flow of the manuscript. Just a suggestion.

Minor Comment 2: IFNg is missing the g in Figure 9.
